# Facile Fabrication of Superwetting PVDF Membrane for Highly Efficient Oil/Water Separation

**DOI:** 10.3390/polym15020327

**Published:** 2023-01-09

**Authors:** Jinzhu Yang, Wei Sun, Junping Ju, Yeqiang Tan, Hua Yuan

**Affiliations:** Key Laboratory of Bio-Fibers and Eco-Textiles, Collaborative Innovation Center of Marine Biobased Fiber and Ecological Textile Technology, Institute of Marine Biobased Materials, College of Materials Science and Engineering, Qingdao University, Qingdao 266071, China

**Keywords:** superhydrophilic, underwater superoleophobic, oil/water separation

## Abstract

A novel superhydrophilic and underwater superoleophobic modified PVDF membrane for oil/water separation was fabricated through a modified blending approach. Pluronic F127 and amphiphilic copolymer P (MMA-AA) were directly blended with PVDF as a hydrophilic polymeric additive to prepare membranes via phase inversion induced by immersion precipitation. Then, the as-prepared microfiltration membranes were annealed at 160 °C for a short time and quenched to room temperature. The resultant membranes exhibited contact angles of hexane larger than 150° no matter whether in an acidic or basic environment. For 1, 2-dichloroethane droplets, the membrane surface showed a change from superoleophilic to superoleophobic under water with aqueous solutions with pH values from 2 to 13. This as-prepared membrane has good mechanical strength and can then be applied for oil and water mixture separation.

## 1. Introduction

A large amount of oily wastewater is produced due to the rapid pace of industrialization, which presents an issue for the survival and development of human society [1]. Several traditional approaches are used to separate oil/water mixtures, including centrifuge and flotation and skimming [2,3]. However, the traditional separation technologies are energy-intensive and require complicated machinery. Thus, membrane materials have been an important separation technology over the past several decades because of their relatively simple operational process, low energy requirement, high stability and good separation efficiency [4,5,6].

Poly (vinylidebe fluoride) (PVDF), due to its excellent chemical stability, thermal stability and radiation resistance [7,8], has been an important microfiltration material. However, the PVDF membrane is easily fouled by organic proteins and other biomolecules, which limits its practical application in the separation process. Surface stable water hydration is generally considered as the key to its resistance to nonspecific protein adsorption [9]. Considerable work, including surface grafting [10], surface coating [11] and blending with hydrophilic polymers, has been done to prevent protein adsorption on microfiltration membrane surfaces to enhance the antifouling ability of the membrane. By comparison, the blending method is the most adapted because of its versatile controlling conditions for preparing a hydrophilic and good fouling resistance membrane [12,13]. Polyvinylpyrrolidone (PVP) and polyethyleneglycol (PEG) [14,15] are the simplest, most widely hydrophilic polymers and can be directly blended with PVDF to improve its anti-fouling property. However, the elution of these additives is unavoidable during the membrane formation and filtration process. Thus, amphiphilic copolymers [16,17,18] have recently been synthesized and used for blending with PVDF to fabricate antifouling microfiltration membranes. The hydrophilicity and fouling resistance of the PVDF membranes using the three amphiphilic polymers as modifiers were better than the membrane using PEG [19]. A novel copolymer with oppositely charged groups was prepared by Shen et al., and was blended with PVDF to resist effective protein adsorption and initial microbial adhesion [20]. Liu et al. developed a porous PVDF hollow fiber membrane in the phase process using the ampliphilic brush-like copolymer P (MMA-EGMA) as the macromolecular additive. The hollow fiber membrane had good protein fouling resistance and the flux was recovered easily by simple water washing [21]. However, there is little in the literature describing a membrane being made hydrophilic by only a blending method to separate oil /water mixture.

Membranes with superoleophobic surfaces could help solve the oil fouling problem and achieve the effective separation of oil/water mixtures [22,23]. Normally, superoleophobic surfaces in air are very hard to attain due to the low surface tension of organic liquids [5]. Hydrophilic surfaces with low values of polymer–water interfacial energy provide stable water layers to restrain protein adsorption and oil adhesion. Membranes with underwater superoleophobicity are a possible method of separating oil/water mixtures without exhausting any external energy, using superhydrophilic and underwater superoleophobic hydrogel-coated mesh for oil/water separation [6,24]. Many kinds of modification methods have been developed to enhance the hydrophilicity of membranes [25,26,27]. However, there are still some limitations for the wide application of surface modification because the complicated process of polymerization is inevitable.

Herein, a simple strategy through a two-step method is used to fabricate a superhydrophilic and underwater superoleophobic PVDF membrane with a hierarchically structured surface. Pluronic F127 and amphiphilic copolymer P (MMA-AA) synthesized as hydrophilic additives were blended with PVDF to attain a modified membrane. Then, the as-prepared microfiltration membranes were annealed at 160 °C for a short time and quenched to room temperature. Thus, a superhydrophilic PVDF membrane was achieved by blending with subsequent thermal treatment. In comparison with the traditional methods, such as surface coating and surface grafting, the preparation process of modified blending is very simple, time-saving and inexpensive. Moreover, the mechanical strength of the membrane after the heating treatment is higher than that of the membrane without the overheating treatment, which is important for a wide range of practical applications. This novel kind of hybrid membrane to separate oil/water mixtures has high water flux and energy-saving filtration under ultralow transmembrane pressure.

## 2. Experimental Section

### 2.1. Materials

The PVDF powders and Pluronic F127 were purchased from Sigma-Aldrich (Shanghai, China). N, N-dimethylacetamide (DMAc), hexane, ethanol, 1, 2-dichloroethane and oil red (sudan III) were used as received. Methyl methacrylate (MMA) and acrylic acid (AA) were obtained from Tianjin Chemical Reagents Company (Tianjin, China).

### 2.2. Synthesis of Amphiphilic Copolymer P (MMA-AA)

In a typical polymerization process, monomers of MMA (8.844 g), AA (16.4223 g) and AIBN (0.088066 g) were dissolved in DMAc. The solution was purged with argon for 30 min and the reaction was at 80 °C for 24 h under a nitrogen atmosphere, afterward precipitating in ethyl ether; the obtained copolymers were dried under vacuum at 30 °C overnight. The synthesized P (MMA-AA) was characterized by ^1^H NMR. Appendix A shows ^1^H NMR spectrum of P (MMA-AA). ^1^H NMR (DMSO, δ, ppm): 0.67–1.02 (-CC*H_3_*), 1.6–2.05 (-C*H_2_*), 3.55 (-OC*H_3_*) and 12.24 (-COO*H*).

### 2.3. Microfiltration Membrane Preparation

Microfitration membrane was prepared via the non-solvent induced phase separation. For PVDF membranes, PVDF (15%) was dissolved in the DMAc by vigorous stirring until a clear homogeneous solution. For the PVDF-F127 composite membranes, PVDF (15%) and F127 (1%) were dissolved in the DMAc. For the PVDF-F127-P (MMA-AA) composite membranes, PVDF (15%) and F127 (1%) were completely dissolved in the DMAc and then the P (MMA-AA) (1%) was added to the solution. Last, the solutions were cast on glass plates by a steel knife and then immersed in a coagulation bath of ethanol. The obtained membranes were kept in deionized water for 12 h and dried in a vacuum oven.

### 2.4. Heating Treatment of As-Prepared Microfiltration Membrane

The three kinds of as-prepared microfiltration membranes were annealed at 160 °C for a short time. After heating treatment, the membranes were quenched to room temperature.

### 2.5. Oil/Water Separation Experiment

The prepared membranes were placed between one vertical glass tube with a diameter of 40 mm and one conical flask. The oil/water mixtures were poured into the filter.

### 2.6. Characterization of Membranesa

Scanning electron microscopy images were obtained with a field emission scanning electron microscope (JSM-6701F, Hitachi, Japan) to analyze the surface structure. The chemical composition of the prepared membrane was obtained by X-ray photoelectron spectroscopy (XPS), which was conducted on a PHI-5702 electron spectrometer (Perkin-Elmer, Waltham, MA, USA). Water contact angle of membrane surface was tested on a DSA100machine (Krüss, Hamburg, Germany). The membrane was placed in a transparent and cubic quartzose vessel filled with ultrapure water first. An oil droplet (1, 2-dichloroethane) was directly placed onto the membrane surfaces. For oils with lower density than water (hexane), the surfaces were first fixed upside in the container, and then an inverted needle was used to place the oil droplet under the surface [28]. The mechanical properties of PVDF membranes were evaluated by tensile tests performed on a Shimadzu AG-X with a strain rate of 2mm/min [29].

## 3. Results and Discussion

Non-solvent induced phase separation (NIPS) is a simple and convenient method to prepare porous membrane. The PVDF membrane precipitated from ethanol, as shown in Figure 1, shows the typical symmetric structure. The cross-section presents a microporous structure composed of spherical particles. It is well known that PVDF is a semicrystalline polymer. Such a structure indicates that the precipitation process is a crystallization-dominated precipitation via nucleation and crystal growth [30]. However, it is believed that the virgin PVDF membrane shows very low tensile strength due to poor adhesion between the spherical particles from the cross-sectional structure. Thus, the tensile strength of the membrane is the first problem to be solved. Quenching is a very simple method to improve membrane tensile strength [31,32]. Take the case of the virgin PVDF membrane: the tensile strength of the membrane and the membrane via the heating treatment are shown in Figure 2A. The mechanical property changes are obvious and the tensile strength of the PVDF membrane via the heating treatment is increased to 2.5 MPa, which is an important determinant for practical application. The first reason is given from the SEM analysis results. The SEM of the virgin PVDF membrane after the heating treatment is also shown in Figure 1c,d, which is different from the untreated PVDF membrane. The surface has the typical agglomerates of interconnected bulge. The spherical particles of the cross-section become small and interconnected. The reason for the morphological change is that at a low quenching temperature the crystallization rate is fast, so more clusters are formed but their augmentation time is short; as a result, the size of the spherical clusters remains small [33]. The thickness of membranes becomes denser after the heating treatment, as shown in Appendix A, which is another reason for the improvement of the tensile strength of the membranes. A low quenching temperature is propitious to obtaining a distinct mechanical property.

The contact angle for water on the virgin PVDF membrane is above 110° and for oil it is 0° (in Figure 3); that is, PVDF is intrinsically hydrophobic and oleophilic due to the low surface energy of PVDF. Therefore, PVDF membranes are easily fouled because of surfactant adsorption or pore plugging by many oil droplets. The hydrophilic improvement of the membrane is the second problem solved. Blending modification is an effective method that can be applied to industrial-scale production. Thus, hydrophilic Pluronic F127 and amphiphilic copolymer P (MMA-AA) are used to blend with PVDF to enhance the surface hydrophilicity of the membranes through forming hydrogen bonds with surrounding water molecules to reconstruct a thin water layer on the membrane surface. However, it is very surprising that the novel composite membrane PVDF-F127 and PVDF-F127-P (MMA-AA) turn superhydrophilic after a simple heating treatment method, as shown in Figure 3, and the surface wettability of the membrane will be discussed in detail below [34].

The SEM of the PVDF-F127 and PVDF-F127-P (MMA-AA) membranes after the heating treatment is shown in Appendix A. The morphology change is analogous to the original PVDF membrane. Meanwhile, the treated PVDF-F127 and PVDF-F127-P (MMA-AA) membranes also show better tensile strength than the untreated membranes, as shown in Figure 2B,C. The reason, as with the PVDF membrane, is that the morphology and thickness of the membranes have changed.

How much can the surface wettability of the membrane have changed? The surface chemical compositions of the pristine PVDF, PVDF-F127 and PVDF-F127-P (MMA-AA) membranes and of all the membranes after the heating treatment were compared via XPS measurements and the results are shown in Figure 4. The XPS spectra of pristine PVDF membranes and membranes after heating treatment (Figure 4 A1–A2) both have two strong peaks at 284.8 eV and 684.7 eV, assigned to C1s and F1s, respectively. In the cases of the PVDF-F127 membrane and the PVDF-F127-P (MMA-AA) membrane after heating treatment (Figure 4 B2–C2), a strong O1s peak is detected at 530.9 eV compared with the untreated membranes (Figure 4 B1–C1), which is the element from the F127 and P (MMA-AA) polymer. The corresponding composition of these membrane surfaces is listed in Figure 4D. It is also noted that the oxygen content increases sharply for the PVDF-F127 and PVDF-F127-P (MMA-AA) membranes after the heating treatment, which confirms the anchoring of the F127 and P (MMA-AA) polymers at the membrane surface [35,36].

The enrichment of F127 and P (MMA-AA) at the membrane surface may be due to the thermodynamic incompatibility between PVDF and the hydrophilic chains of F127 and P (MMA-AA) during the heating treatment process. The hydrophobic chains (PPO and PMMA segments) can guarantee the compatibility of PVDF with hydrophobic materials, while the hydrophilic PEO and PAA chains stretch out of the membrane surface during the heating treatment process due to the segregation impact, providing a high coverage of a hydrated polymer layer anchored by a hydrophobic backbone that is not soluble in water and entangled with the PVDF polymer [37].

The C1s core-level spectrum of the pristine PVDF membrane shows two typical peaks in Figure 5A, one at a binding energy (BE) of 284.8 eV for the carbon bonded to hydrogen (CH_2_) and the other at a BE of 289.2 eV for the carbon single bonder to flour (CF_2_). Concerning the PVDF-F127 membrane, as shown in Figure 5B, the C1s core-level spectrum can be resolved into seven representative peaks corresponding to CF_2_ (289.2 eV) and CH_2_ (284.8 eV) assigned to PVDF, CH (283.4 eV), C-O (285.0 eV PEG) assigned to F127. Concerning the PVDF-F127-P (MMA-AA) membrane, three other new peak components (with a BE at 287.4 eV for the COO species, at 284.1 eV for the C-COO species and at 285.2 eV for the COO-C species) have appeared in the C1s core-level spectrum (Figure 5C). Furthermore, the peak component area ratio for [C-COO]/[COO-C]/[COO] is about 5:3:5 according to the integral area, which is in good agreement with the theoretical ratio for the chemical structure of P (MMA-AA). Based on the above results, the surface chemical compositions of the membranes are changed by intruding hydrophilic polymers together with a simple heating treatment to attain a superhydrophilic membrane.

In this section, the surface wettability of the membrane was evaluated by the contact angle measurements, as shown in Figure 6. According to the Wenzel mode, for a hydrophilic substrate, the superhydrophilic surface can be attained by enhancing surface roughness due to the capillary effect. Analogously, an oleophilic surface will become more superoleophilic. The pristine PVDF membrane shows hydrophobic and superoleophilic properties, which are attributed to the micro/nanohierarchical rough structure arising from phase separate technology together with the high surface free energy. Thus, the pristine PVDF membrane is not suited to separate oil/water mixtures. For the PVDF-F127 membrane, the water droplet can spread out and permeates through the membrane immediately, exhibiting a superhydrophilic character, which is also due to the micro/nanohierarchical rough structure and the hydrophilic chain enrichment from F127 on the membrane surface. Figure 6a,b show the wetting behavior of underwater oil (hexane and 1, 2-dichloroethane) on the membrane with increasing water pH. When a hexane droplet makes contact with the membrane surface, oil exhibits a high CA above 130° and the contact angles remain constant in acidic and even basic water (pH 2–12). The reason is that when the PVDF-F127 membrane is immersed in water, water is trapped into the hierarchical structure to form an oil/water/solid interface in the presence of the oil and the trapped water serves as a support to prevent the penetration of the oil droplets. However, when a 1, 2-dichloroethane droplet makes contact with the membrane surface underwater, it immediately spreads out and a contact angle of nearly 0 is observed in acidic and even basic water (pH 2–12). It is hard to explain this uncommon wetting phenomenon on the membrane surface by using the Cassie theory and Wenzel. It seems a great challenge to explain this unconventional wetting phenomenon by utilizing a strategy for tunable surface wettability through the control of the Lewis acid-base interactions at the liquid–liquid interface, through which a tunable liquid–liquid interfacial tension can be achieved [38]. The possible reason is that dichloromethane has a much greater density than water and the layer of trapped water is very thin. An oil droplet is easy to spread out on the membrane surface, so an underwater superoleophilic phenomenon is attained and the PVDF-F127 membrane is also not very suited to separate oil–water mixtures.

The underwater oil contact angles on the PVDF-F127-P (MMA-AA) membrane surface as a function of the pH value in the aqueous phase are also shown in Figure 6c,d. It is shown in Figure 6c that the contact angles of hexane are larger than 150° no matter whether the aqueous phase is acidic or basic. The variation in the oil contact angles is in good agreement with the wetting of the PVDF-F127 membrane. For the 1, 2-dichloroethane droplet, as shown in Figure 6d, the contact angle is 0° and keeps constant with aqueous solutions with pH values from 2 to 9, indicating the surface shows superoleophilicity. While the aqueous phase changes to basic (pH > 10), the 1, 2-dichloroethane droplet shows a spherical shape with a contact angle larger than 150°. The membrane surface displays a change from superoleophilic to superoleophobic underwater with aqueous solutions with pH values from 2 to 13. Considering the membrane surface is pH responsive, it is supposed that the surface wetting transition can be attributed to the surface chemical composition changes on the substrates. When the pH values are over 10, the membrane surface acid carboxyl groups dissociate into carboxylate ions, leading to the enhancement of hydrogen bonds between the membrane and water. In addition, with the presence of hierarchical structures, water molecules near the membrane surface region more easily form a continuous, compact and stable hydration layer, which can exert a strong repulsive force to oil trying to approach the membrane surface. Thus, it is expected that 1, 2-dichloroethane shows a macroscopic CA exceeding 150° in a basic aqueous environment. Figure 7 shows a schematic illustration of the change of the PVDF-F127-P (MMA-AA) membrane after immersion in a basic environment to present the change of the membrane surface charge.

Figure 8 shows the CA of the 1, 2-dichloroethane droplets on the PVDF-F127-P (MMA-AA) membrane surface with a pH from 12 to 2. When the pH values are larger than 5, the surfaces are superoleophobic underwater with an oil CA above 150°. For the oils in an acidic aqueous environment (pH < 5), the surface shows underwater superoleophilicity. The pKa of PAA in solution is approximately 4.3–4.9. When the pH is lower than 5, the carboxyl acid groups of the PAA chains are protonated and could lead to weak hydrogen bonds forming between the acid carboxyl groups of PAA and water. The result is that the membrane is immersed in a basic environment before use and can be used to separate the oil–water mixtures at pH > 5.

The oil/water separation capability of PVDF-F127-P (MMA-AA) membranes was performed as shown in Figure 9. In the case of the PVDF-F127-P (MMA-AA) membrane, a 200 mL mixture of hexane and water, where hexane was dyed red using oil Red O, was poured onto the as-prepared membrane. As shown in Figure 9A, the water including acidic and basic water permeated through the membrane within 3 min and oil was retained above due to the underwater superoleophobicity and low oil-adhesion properties. In addition, as shown Figure 9B, hexane cannot permeate through the filter membrane even after 24 h, which indicates this oil-removing membrane is a good candidate in industrial oil-polluted water treatments.

## 4. Conclusions

Pluronic F127 and amphiphilic copolymer P(MMA-AA) as hydrophilic polymer additives were used for blending with PVDF to attain a modified membrane. Then, a superhydrophilic membrane was achieved when the as-prepared microfiltration membrane was annealed at 160 °C for a short time and quenched to room temperature. The membranes show good mechanical strength. The PVDF-F127-P(MMA-AA) membrane shows underwater superoleophobic and low oil-adhesion properties that allow for the effective separation of oil–water mixtures. It is expected to be a promising candidate for applications in industrial oil-polluted water treatments and oil spill cleanup.

## Figures and Tables

**Figure 1 polymers-15-00327-f001:**
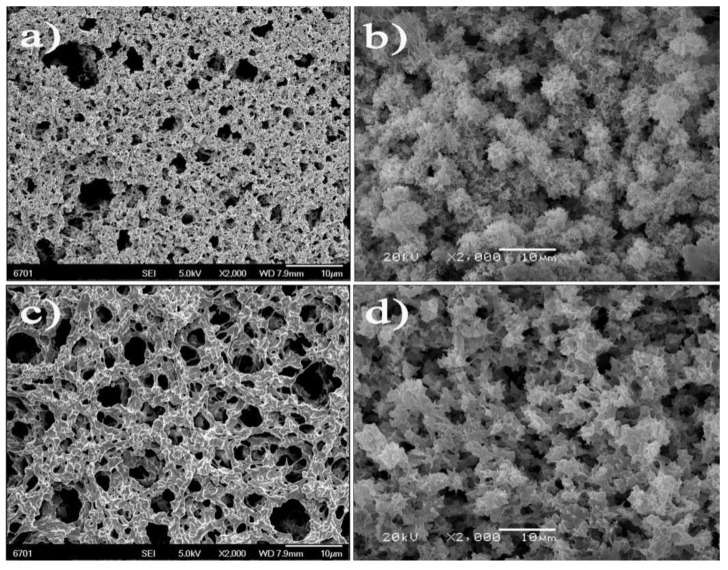
(**a**,**b**) surface and cross-sectional structure images of the original PVDF membrane. (**c**,**d**) surface and cross-sectional structure images of the original PVDF membrane after heating treatment.

**Figure 2 polymers-15-00327-f002:**
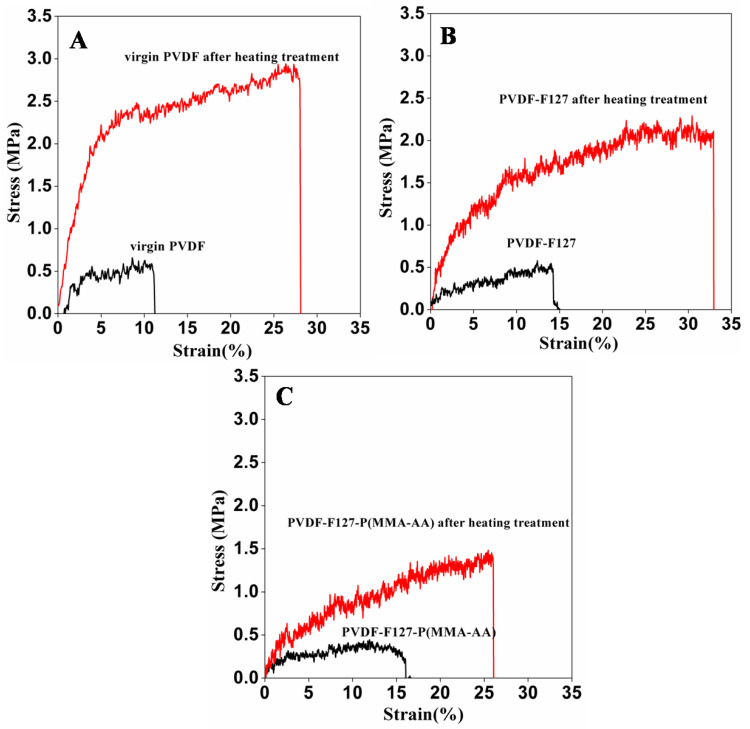
Stress-strain diagrams for PVDF, PVDF-F127, PVDF-F127-P (MMA-AA) membranes. (**A**) PVDF; (**B**) PVDF-F127; (**C**) PVDF-F127-P (MMA-AA).

**Figure 3 polymers-15-00327-f003:**
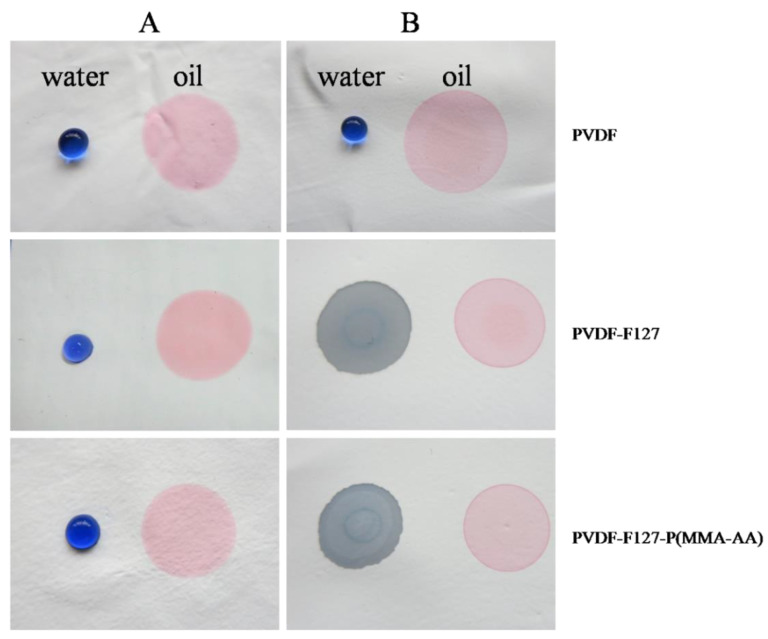
Droplets of water (dyed blue) and n-hexane oil (dyed red) on the PVDF, PVDF-F127, PVDF-F127-P (MMA-AA) membranes: (**A**) without heating treatment, (**B**) via heating treatment, respectively.

**Figure 4 polymers-15-00327-f004:**
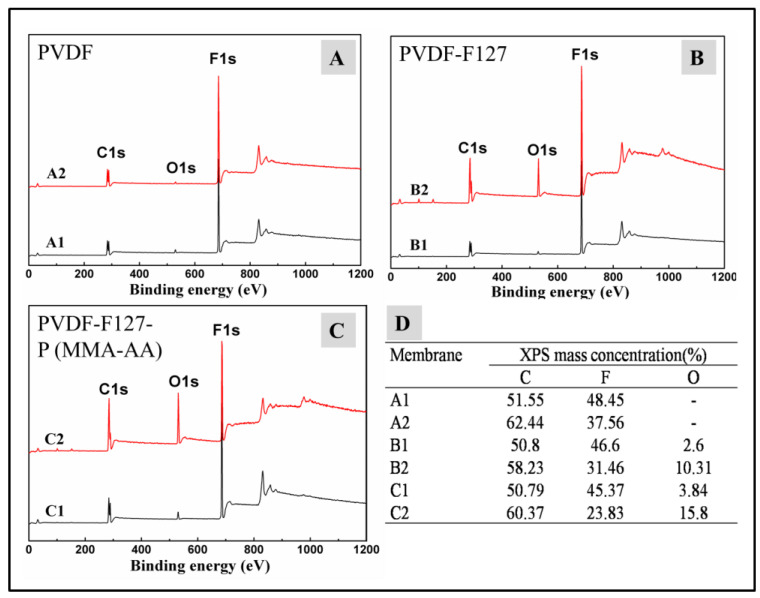
The XPS wide scan spectra of (**A**) PVDF; (**B**) PVDF-F127; (**C**) PVDF-F127-P (MMA-AA): (A1, B1, C1) membranes before heating treatment; (A2, B2, C2) membranes after heating treatment, respectively. (**D**) Surface element and composition of membranes.

**Figure 5 polymers-15-00327-f005:**
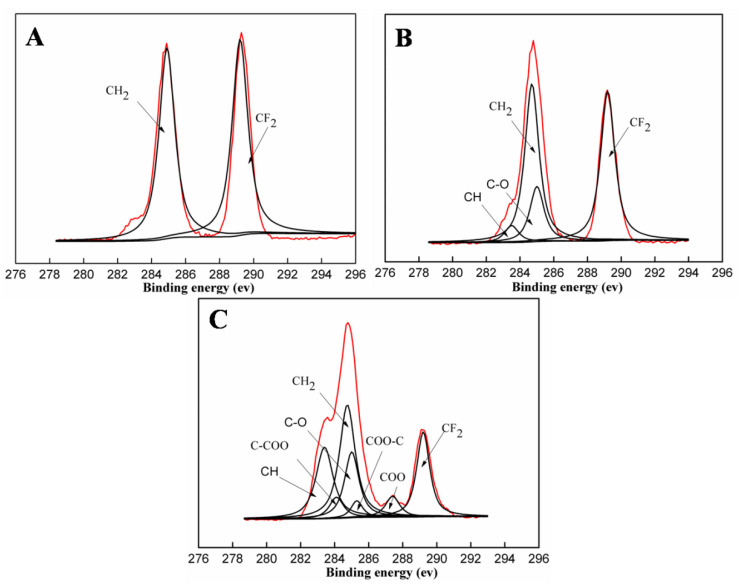
The C1s core level spectra of membranes after heating treatment: (**A**) PVDF, (**B**) PVDF-F127 and (**C**) PVDF-F127-P (MMA-AA).

**Figure 6 polymers-15-00327-f006:**
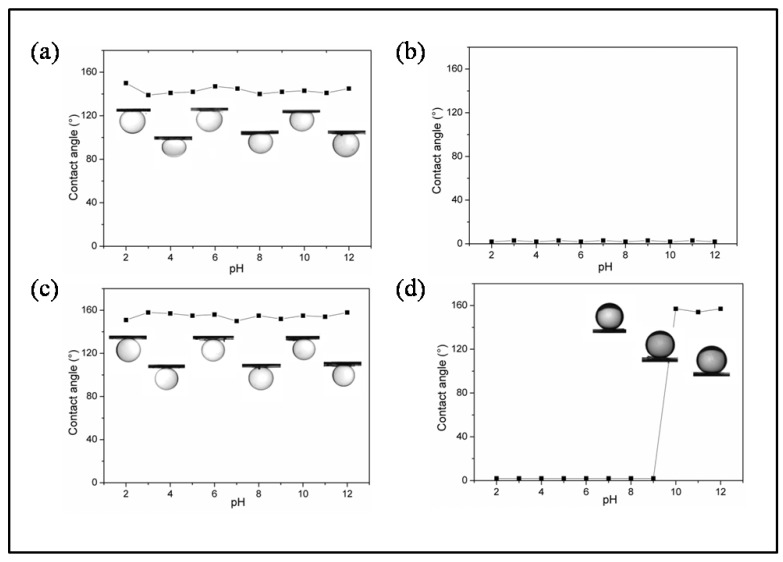
Contact angles (CA) of the oil droplets on the membrane as a function of the pH value in the aqueous phase: (**a**) hexane and (**b**) 1, 2-dichloroethane on PVDF-F127 membrane surface, respectively; (**c**) hexane and (**d**) 1, 2-dichloroethane on PVDF-F127-P (MMA-AA) membrane surface, respectively.

**Figure 7 polymers-15-00327-f007:**
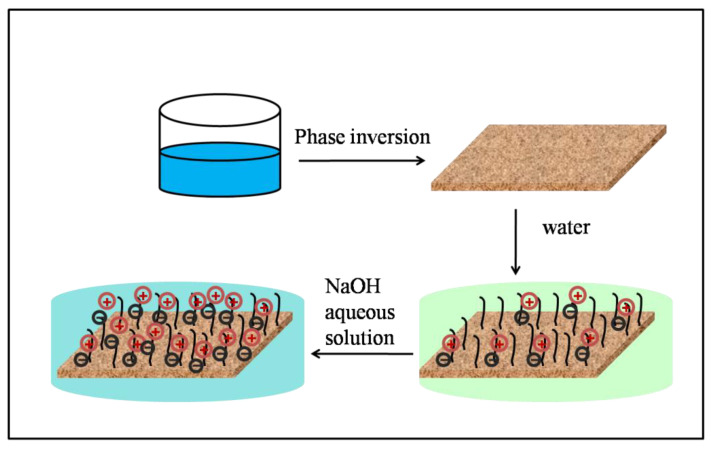
Schematic illustration of the change of PVDF-F127-P (MMA-AA) membrane after immersion in basic environment.

**Figure 8 polymers-15-00327-f008:**
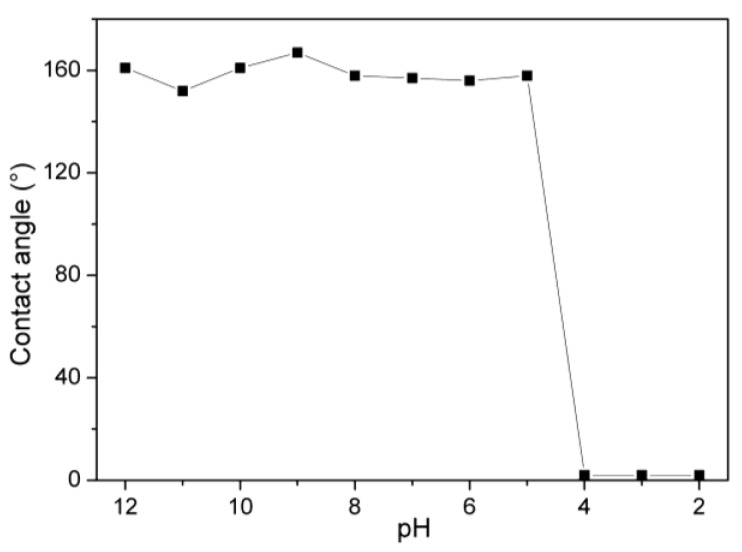
Contact angles (CA) of 1, 2-dichloroethane on PVDF-F127-P (MMA-AA) membrane surface as a function of the pH value.

**Figure 9 polymers-15-00327-f009:**
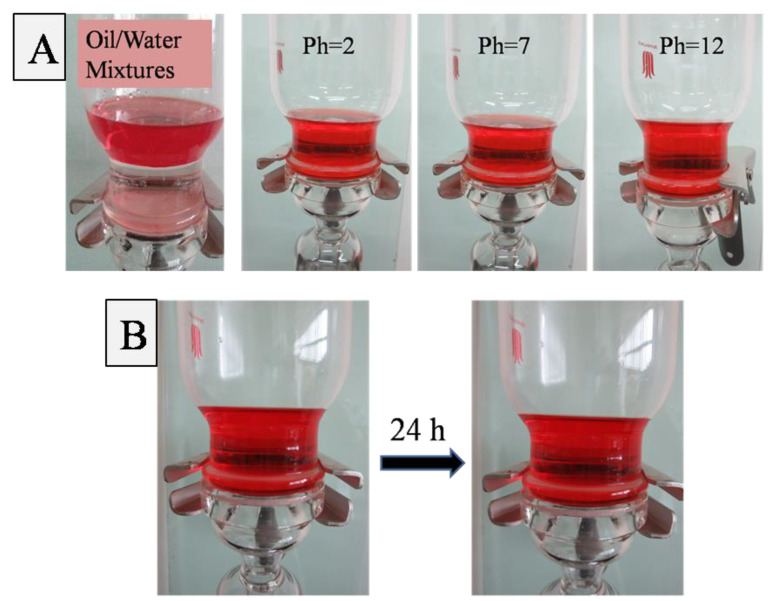
Oil/water separation experiment for PVDF-F127-P (MMA-AA) membrane: (**A**) the water including acidic and basic water permeating through the membrane; (**B**) hexadecane permeating through membrane after 24 h.

## Data Availability

Not applicable.

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
