# Peer review of "Facile Fabrication of Superwetting PVDF Membrane for Highly Efficient Oil/Water Separation"

_polymers, 2023, doi:10.3390/polym15020327_

Round 1

Reviewer 1 Report

The review of the paper "Facial strategy for preparation of modified PVDF membrane: endowing effective separation of oil/water mixtures" 

I do not understand why you titled this paper "strategy". It is a standard paper about membrane preparation.

Some references should support the discussion. There are many papers about Pluronic impact on membrane properties. Please consider citing this papers:

DOI:10.5004/dwt.2021.26647

https://doi.org/10.1016/j.memsci.2007.02.011

https://doi.org/10.1016/j.memsci.2008.03.013

Author Response

Response to Reviewer 1 Comments

We are grateful for your comments and criticism. Indeed, your comments are very helpful for refining the original manuscript to a revised one. We took the following actions to your comments. We hope if these changes would be sufficient for publication of this paper in Polymers.

Point 1: The review of the paper "Facial strategy for preparation of modified PVDF membrane: endowing effective separation of oil/water mixtures"

I do not understand why you titled this paper "strategy". It is a standard paper about membrane preparation.

Response 1: Many thanks for the reviewer’s comment. We have assessed our manuscript to some degree and all authors are favourable to your idea. Now, the title of the revised manuscript is “Facile fabrication of superwetting PVDF membrane for highly efficient oil/water separation”.

Point 2: Some references should support the discussion. There are many papers about Pluronic impact on membrane properties. Please consider citing this papers: DOI:10.5004/dwt.2021.26647

https://doi.org/10.1016/j.memsci.2007.02.011, https://doi.org/10.1016/j.memsci.2008.03.013.

Response 2: Thanks for the reviewer for providing us these valuable references. Both of these literatures reported the pluronic impact on membrane properties, which were closely related to our current work, and have large guiding significances. We have cited these works in the revised manuscript (DOI:10.5004/dwt.2021.26647; DOI:10.1016/j.memsci.2007.02.011; DOI: 10.1016/j.memsci.2008.03.013).

Reviewer 2 Report

Recommendation: Publish after minor revisions noted.

This work concentrates on the preparations of a novel superhydrophilic and underwater superoleophobic modified PVDF membrane for oil/water separation. Experimental results indicate that this as-prepared membrane has good mechanical strength and can then be applied for oil and water mixture separation. Here, after going through your paper, I have some comments for you in order to improve your work.

1. The abstract should be concise and state the highlights of this manuscript.

2. Please add a space between the value and its unit. Check throughout the draft.

3. The grammar should be carefully revised before the manuscript is published.

Author Response

This work concentrates on the preparations of a novel superhydrophilic and underwater superoleophobic modified PVDF membrane for oil/water separation. Experimental results indicate that this as-prepared membrane has good mechanical strength and can then be applied for oil and water mixture separation. Here, after going through your paper, I have some comments for you in order to improve your work.

Point 1: The abstract should be concise and state the highlights of this manuscript.

Response 1: Thanks for the reviewer’s suggestion. The abstract has been concised and stated the highlights of this manuscript.

Abstract: A novel superhydrophilic and underwater superoleophobic modified PVDF membrane for oil/water separation was fabricated through a modified blending approach. Pluronic F127 and amphiphilic copolymer P (MMA-AA) were directly blended with PVDF as hydrophilic polymeric additive to prepare membranes via phase inversion induced by immersion precipitation. Then, the as-prepared microfiltration membranes were annealed at 160 ℃ for a short time and quenched by room. The resultant membranes exhibited that the contact angles of hexane are larger than 150 ° no matter in acidic or basic environment. For 1, 2-dichloroethane droplet, the membrane surface showed a change from superoleophilic to superoleophobic under water with aqueous solutions with pH values from 2 to 13. This as-prepared membrane has good mechanical strength and can then be applied for oil and water mixture separation.

Point 2: Please add a space between the value and its unit. Check throughout the draft.

Response 2: Thanks for the reviewer’s suggestion. We have checked the manuscript and add a space between the value and its unit.

Point 3: The grammar should be carefully revised before the manuscript is published.

Response 3: We are very sorry for our poor English language in our manuscript. Special thanks to you for your good comments. Many basic grammar and other mistakes throughout the manuscript are seriously examined and corrected in the revised manuscript.

Round 2

Reviewer 1 Report

The authors provided some changes to the paper. I am glad to reccomend the paper to publication.